# Longitudinal Study of the Mental Health, Resilience, and Post-Traumatic Stress of Senior Nursing Students to Nursing Graduates during the COVID-19 Pandemic

**DOI:** 10.3390/ijerph192013100

**Published:** 2022-10-12

**Authors:** Ana Isabel Cobo-Cuenca, Beatriz Fernández-Fernández, Juan Manuel Carmona-Torres, Diana P. Pozuelo-Carrascosa, José Alberto Laredo-Aguilera, Benjamín Romero-Gómez, Sergio Rodríguez-Cañamero, Esperanza Barroso-Corroto, Esmeralda Santacruz-Salas

**Affiliations:** 1Facultad de Fisioterapia y Enfermería, Grupo IMCU, Departamento de Enfermería, Fisioterapia y Terapia Ocupacional, Universidad de Castilla-La Mancha, Campus de Fábrica de Armas, Av de Carlos III s/n, 45071 Toledo, Spain; 2Hospital General Nuestra Sra. Del Prado, Servicio de Salud de Castilla-La Mancha (SESCAM), Av. Extremadura KM 114, 45600 Talavera de la Reina, Spain; 3Facultad de Enfermería, Grupo IMCU, Universidad de Castilla-La Mancha, Santa Teresa Jornet s/n., 16071 Cuenca, Spain; 4Hospital El Tomillar de Sevilla, Servicio Andaluz de Salud (SAS), 41500 Alcalá de Guadaira, Spain; 5Clínica Hemodiálisis Avericum Toledo, Calle Alemania 141, 45005 Toledo, Spain

**Keywords:** COVID-19, anxiety, coping strategies, health, students, nursing

## Abstract

This study analyzed changes in the psychological health of students who were in the final year of their nursing degree during the COVID-19 pandemic and later served as nursing professionals in hospitals. Methods: A prospective longitudinal study was conducted over two periods of time (the first in April 2020 and the second 6 months later, in December 2020) with 296 students for a T0 baseline (rate response 68.83%) and 92 students for a T1 post-test sample (response rate 31.08%). The data were electronically collected using the Hospital Anxiety and Depression Scale, the Life Satisfaction Questionnaire, the Resilience Scale, and a post-traumatic stress questionnaire. The mean age of the sample participants was 24.17 years (SD = 5.51), and 89.11% were female. During the pandemic, 14.11% of students showed scores that indicated depression, and 32.61% showed scores that indicated anxiety. In December 2020, 86.5% of the participants were working as nurses, and the percentages of those with anxiety (12%) and depression (4.3%) were significantly lower than in the first sample period. A total of 20.7% of the participants had post-traumatic stress. High scores for resilience were significantly associated with better quality of life and lower levels of anxiety, depression, and post-traumatic stress. Conclusions: Although the percentages of participants with anxiety and depression decreased, they still presented with mental health problems.

## 1. Introduction

In December 2019, the first case of pneumonia originating from the SARS-CoV-2 virus was identified in the city of Wuhan, China. The virus was named according to its clinical similarity to the SARS virus that was identified in 2003 and determined to be responsible for the COVID-19 disease [1]. The virus’s great capacity for infection and propagation led the World Health Organization to declare COVID-19 a pandemic on 11 March 2020 [2].

Due to this increase in the number of cases, on 14 March 2020, a state of alarm was decreed for the entire national territory of Spain, and the confinement of the population was mandated. The declaration of a state of alarm places all powers in the hands of the central government with the objective of mobilizing all available means to protect the state’s health resources [3]. Despite all the measures that were adopted, the total number of diagnosed cases and deaths in the first weeks of the state of alarm was very high, as was the proportion of health personnel who were infected due to the lack of material resources, such as masks and personal protective equipment (PPE) [4,5]. In fact, at the end of May 2020, Spain (Madrid, Castile-La Mancha, and Castile-Leon) and Italy (Lombardy) were the regions with the highest excess mortality rates from COVID-19, both of which were approximately 30% [6]. A report issued by the National Institute of Statistics of Spain on 15 November 2021 showed that the COVID disease was the main cause of death in Spain in 2020, with its highest impact in the months of March and April (with 60,358 deaths and a deathrate of 127.5 per 100,000 inhabitants) [7]. Of the confirmed cases, 26% were health professionals, and Spain was the country with the highest number of health professionals infected by COVID-19 [8].

In terms of the impacts of COVID-19 on education, efforts to reduce the spread of COVID-19 through social isolation and mandatory social immobilization led to the closure of higher education institutions. University education was no longer provided via in-person classroom instruction; COVID-19 forced a change in the learning structure in university education and in the ways of teaching, with technologies playing a significant role [9]. At the University of Castilla-La Mancha (UCLM), the rector suspended in-person learning by enforcing the resolution of 11 March 2020, which adopted preventive measures and public health recommendations related to the university community as a result of issues surrounding the situation and evolution of COVID-19 [10]. At that time, the clinical practice of nursing students was interrupted. Due to the lack of health personnel and the overflow of patients in the health system, on 27 March 2020 [11], the Spanish government adopted measures to strengthen the health system. These measures included the authorization of students in their last year of nursing school to work at health centers as health assistants. Many of fourth-year students volunteered to provide health care in the absence of health personnel.

Public health emergencies can have many psychological effects on university students, including anxiety, fear, and worry [12]. A study of 52,730 people in China conducted during the initial phase of the pandemic found that 35% of the participants experienced psychological stress, with higher levels among those aged 18 to 30 years and, particularly, among women. Another study conducted in China revealed that 24.9% of university students suffered anxiety due to the COVID-19 outbreak [12]. The anxiety of university students as a result of COVID-19 may have been related to the effect of the virus on their studies, their use of social networks as their main means of information [13], and their fears of contagion [13], social distancing, and social alarmism [14], as well as anxiety surrounding their future job prospects [15]. These results indicate that the COVID-19 crisis could have had a significant psychological impact on university students. For nursing students, this impact would have been greater. In Spain, during the first wave, many nursing students had contact with COVID-19 patients during clinical practice, some contracted the disease, and many final-year students provided health care. Several studies explored the psychological and mental health impacts of the first wave of the COVID-19 epidemic on nursing students, showing an increase in fear, anxiety, depression, and sleep problems [16,17].

Similar data were found with healthcare professionals during the first pandemic period, with studies reporting a high incidence of anxiety, depression, and post-traumatic stress among participants [18,19,20]. The impact of COVID-19 was not equal among all healthcare professionals: gender (female), profession (nurse), and shift work increased the incidences of anxiety, depression, and post-traumatic stress. Despite this, resilience has also been considered a protective factor, which would be the person’s ability to confront adversities and thereby reduce anxious, depressive, post-traumatic stress, and burnout symptoms [18,19,20].

After the first wave, longitudinal studies were conducted on the general population [21,22,23]. Luceño et al. (2021) [24] found that the levels of post-traumatic stress, anxiety, and depression of healthcare professionals was reduced with time, although there was not a full recovery.

However, to our knowledge, there are no known longitudinal studies that have evaluated mental health in students who were in their final year of study (fourth-year students in Spain) who have since graduated and are working as nurses.

Therefore, our starting hypothesis for this research was that university nursing students experienced a psychological impact from COVID-19 and may be suffering from anxiety/stress, which can last for several months.

The aims of this study were to analyze the psychological impacts of the COVID-19 pandemic on students who were in the last year of their nursing degree (and who provided health care during that period) and to assess their subsequent psychological health after they graduated from nursing school and began working as nurses in hospitals in Spain. The second objective was to determine the relationships between the different variables and mental health.

## 2. Materials and Methods

### 2.1. Design

This was a prospective longitudinal cohort study with 2 data collection periods (T0: from April to May 2020 and T1: December 2020) with students in the nursing program at the University of X. The Strengthening the Reporting of Observational Studies in Epidemiology (STROBE) guideline has been followed.

The inclusion criteria included:Enrolment in the fourth year of the nursing degree program at the University of X during the 2019/2020 academic year.

The exclusion criteria included:Students having abandoned their studies during the 2019/20 academic year.

To calculate the sample size, the prevalence of anxiety (24.9%) among university students in China was used [12]. The reference population was estimated to be 400 fourth-year undergraduate students at the University of X [25].

The sample size was calculated with the GRANMO program (v 7.12 April 2012) to determine an estimate with respect to a reference (proportional). Accepting an alpha risk of 0.05 and a beta risk of 0.2 in a two-tailed comparison, 45 subjects were required to detect a difference equal to or greater than 0.2 units. It was assumed that the proportion of the reference group was 24.9%. A loss-to-follow-up rate of 10% was estimated.

The questionnaire was sent in 2 stages to all fourth-year nursing students at the University of X. During the first stage (T0), in which all participants were still nursing students, 296 responses were received (the pretest response rate was 68.83%); for the follow-up survey, which was administered 6 months later, 92 responses were received (T0 baseline–T1 post-test response rate of 31.08% (N = 296 students who responded to the first survey) and T0–T1 test response rate of 22.5% (N = 400 enrolled students)).

### 2.2. Variables

-The independent variables were age, gender, smoking status (yes/no), number of cigarettes smoked, and previous pathologies.-The dependent variables were:
o anxiety and depressiono satisfaction with lifeo resilienceo post-traumatic stress

### 2.3. Instruments

The Hospital Anxiety and Depression Scale [26], validated in Spanish by Herrera et al. [27], is a self-administered scale with 2 subscales, each of which includes 7 items (14 items in total). Scores of between 0 and 7 on each scale indicate no concerns regarding anxiety/depression, scores of between 8 and 10 indicate possible anxiety/depression, and scores of above 11 indicate probably anxiety/depression. The scale has an internal consistency by Cronbach’s alpha of 0.90 for the full scale, 0.85 for the anxiety scale, and 0.84 for the depression scale.LISAT-8, the Spanish version of the LISAT-8 [28], is a self-administered questionnaire consisting of 8 items rated on a Likert scale. The items measure the respondent’s satisfaction with 8 different aspects of life (life, sexual life, relationship with their partner, family life, relationship with friends and acquaintances, leisure, work situation, and financial situation). The final score is calculated by adding the scores of all the items (total possible scores range from 8 to 48). The scale comprises 3 dimensions: satisfaction with social life (items 1, 4, 5, and 6), satisfaction with emotional life (items 2 and 3) and satisfaction with work or financial life (items 7 and 8). The scale has a sensitivity of 81.7% (CI: 80.5–82.9) and a reliability of 79.2% (CI: 77.5–80.8) [19].The 10-item short version of the CD-RISC (2003) [29] proposed by Campbell and Stein (2003) [30], adapted to the university population by Notario et al. (2011) [31], was used. This is a self-administered questionnaire comprised of 10 items that uses a Likert scale for answers, with 5 response options ranging from 0 (never) to 4 (almost always). The final score of the questionnaire is the sum of the responses obtained for each item (range of 0–40). A higher score indicates a higher level of resilience. The Cronbach’s alpha was 0.85 and the test-retest intraclass correlation coefficient was 0.71.The short version of the Davidson Trauma Scale (DTS) by Connor and Davidson (2000) [32], adapted to Spanish by Bobes et al. (2000) [33], is a questionnaire comprising 8 items answered using a 5-point Likert scale (0–4 points). It quantifies the frequency and severity of each symptom of post-traumatic stress disorder and the results in a total score of between 0 and 32 points. The cut-off scores proposed by the authors were as follows: 0–7, no post-traumatic stress disorder, and 12 or higher: post-traumatic stress disorder. The Cronbach’s alpha was 0.83.

### 2.4. Procedure

An online questionnaire was developed. Five faculty members at the University of X were contacted and asked to pass the questionnaire link to their fourth-year students.

The questionnaire was distributed at 2 time points:-First data collection period (T0): the HADS questionnaire was administered from April to May 2020 to determine the psychological state of nursing students during social isolation and confinement.-Second data collection period (T1): All the students who completed the online questionnaire during phase 1 were invited to participate in the second phase of the study, which aimed to determine the evolution of the psychological impacts of COVID-19. This assessment was performed in December 2020. In this phase, all the participants now graduated nurses. The questionnaires used were HADS, LISAT 8, CD-RISC, and DTS.

### 2.5. Data Analysis

For the statistical analysis, SPSS version 22 (IBM Corp. Armonk, NY, USA), licensed by the University of Castilla-La Mancha, was used. Qualitative variables are reported as counts and percentages. Quantitative variables are expressed as the arithmetic means (m) and standard deviations (SD). An inferential analysis was performed to identify the relationships between the independent variables and the dependent variables, as follows:For qualitative variables: proportions of categorical variables were compared using χ^2^ tests for contingency tables; for 2 × 2 tables, the χ^2^ statistics with a Yates correction was used, and when the expected frequency was ≤5, Fisher’s exact test was applied.For quantitative variables: first, the goodness of fit to a normal distribution was determined using the Shapiro–Wilk test, and then the homogeneity of the variances was determined using the Levene test. As the data did not follow a normal distribution, we used the appropriate nonparametric (Mann–Whitney U) test.

We used Spearman’s correlation analysis (Rho) to explore the relationships between the different dependent variables (anxiety, depression, satisfaction with life, resilience and post-traumatic stress). In addition, to control the influence of gender, a partial correlation analysis was performed.

To identify the differences in anxiety and depression in the pre-test with those of the post-test, the Wilcoxon rank test was used.

All hypothesis comparisons were two-tailed. For all statistical tests, significant values were those with a *p* value of <0.05 at a confidence interval of 95%.

### 2.6. Ethical Considerations

The present study was approved by the Clinical Research Ethics Committee of the Health Area of X with the code 24/2020. The research respected the fundamental principles of the Declaration of Helsinki of the European Convention on Human Rights and Biomedicine. All participant data were treated confidentially in accordance with Organic Law 3/2018 of December 5 on the Protection of Personal Data and Guarantee of Digital Rights, maintaining strict confidentiality and preventing access to unauthorized third parties. All participants read the information sheet and gave their consent to participate in the study.

## 3. Results

During the first administration (T0), which took place during the confinement period, 296 students (response rate of 68.83%) aged between 21 and 54 years (m = 23.4; SD = 4.79) answered the questionnaires in full.

A total of 92 students participated in both data collection phases (T0 and T1). All were fourth-year students at the time of their inclusion in the study, and 86.5% were working as nurses during the second data collection period. A total of 89.1% were women and 10.9% were men; their ages were between 21 and 54 years (mean = 24.17; SD = 5.51). The majority (76.1%) had no previous pathology. A total of 5.4% were previously diagnosed with anxiety, 2.2% had a previous diagnosis of asthma, 2.2% had previous diagnoses of hypertension and dyslipidemia, and 1.1% had a previous diagnosis of hypothyroidism. With respect to tobacco, only 13 people smoked, and the number of cigarettes they smoked did not change (T0–T1).

Regarding the pre-test phase T0 (May 2020), 47.8% of the participants had provided health care assistance to the public health system in Spain due to the pandemic situation. Of these, 25.60% worked in a call center, 7.41% in intensive care units, and the rest in hospitalization units. With respect to shifts, 74.41% worked rotating shifts, and while doing so, 69.8% always had access to protective material against COVID-19 and only 3.5% underwent testing for COVID-19. During this phase (T0) (April–May 2020), 32.6% suffered from anxiety and 14.1% experienced depression.

Regarding coping with problems during the confinement period (April–May 2020), 51.1% of the respondents performed physical activity several times per week (21.1% were active every day), 35.6% talked to friends every day, 35.6% watched movies or TV shows to distract themselves every day, 34.4% tried to distract themselves with things they liked to avoid thinking about anything several times per week, 22.2% were angrier than usual several times week, 27.8% were more sad than usual several times per week, and 18.9% cried several times per week.

Regarding coping strategies and anxiety and depression during T0, talking with friends every day was associated with less depression (χ^2^ = 9.80, *p* = 0.020), and engaging in physical activity every day or several times per week was associated with less depression (χ^2^ = 10.69, *p* = 0.03). Being angrier than usual several times per week was related to more anxiety (χ^2^ = 26.88, *p* < 0.001) and crying more than usual several times per week was related to anxiety (χ^2^ = 26.56, *p* < 0.001).

During the T1 phase (December 2020), 86.5% of the participants were working as nurses. A total of 20.7% of the participants had post-traumatic stress, while 55.1% considered their quality of life to be high. Finally, the respondents had a mean resilience score of 27.03 (SD ± 8.27). Their responses to the different scales are shown in Table 1.

Table 2 presents the differences in anxiety and depression between the two study phases. There were significant differences because the percentages of anxiety and depression decreased during the post-test phase.

Table 3 shows the correlations among the scores on the different scales during T1. Anxiety was positively associated with depression and post-traumatic stress. Depression was positively associated with post-traumatic stress. However, depression was negatively associated with resilience and the LISAT-8 scores. The LISAT-8 scores were positively associated with resilience and negatively associated with post-traumatic stress. When we conducted the sex-adjusted partial correlation, the results were similar, except for depression, which was not significantly associated with resilience and the LISAT-8 scores.

## 4. Discussion

The aim of this study was to analyze the psychological health of students who were in the last year of their nursing degree during the COVID-19 pandemic and their subsequent psychological health after they graduated from nursing school and began working as nurses in hospitals in Spain.

Many studies have evaluated the impact of the COVID-19 pandemic on the mental health of the general population, health personnel, and nursing students [25,26,27]. However, to our knowledge, this impact has not been longitudinally evaluated in nursing students who were providing health care assistance during the first wave of COVID-19 (March–May 2020) and again 6 months later, when they had graduated and were working as nurses (December 2020).

During the first wave of the pandemic, most university students had to stop attending classes in person. However, in Spain, students in their last year of nursing school had the opportunity to develop health care skills due to the collapse of the health system [11], which created the need for them to serve as front-line health workers in April and May without having finished their studies. This first wave of the pandemic was characterized by a very high mortality rate and little knowledge of the disease [7,8]. In our study, we observed that during the first wave, 47.8% of the participants provided health care assistance to the public health system in Spain prior to the completion of their studies.

In the first wave in Spain, during the confinement period, more than half of the students had symptoms of anxiety, and one-third of the total sample scored above the cut-off point for a diagnosis of anxiety disorder. Additionally, one-third of the participants scored high on the depression scale and 14% met the criteria for depressive disorder. The levels of both anxiety and depression were not related to age, whether the participants had provided health care assistance to the public, work locations, or work shifts. This increase in anxiety and depression during confinement coincided with other studies of nursing students, such as those by Patelarou et al. (2021) [16] and Romero et al. [17], and with other studies of health professionals [18,20]. In other studies [18], anxiety and depression were related to age, with the youngest participants having the highest scores. In our study, this was not the case, possibly because the students in our study were of similar ages and had not previously worked in the health system; they only had experience in clinical practice. They may also have had less fear of being infected [34] or of the repercussions of the disease, which caused higher mortality in older people [7,8].

Another fact that is not consistent with some previous studies [18,20,24,35] is the absence of a significant relationship between gender, anxiety, and depression, which is atypical and could be due to the presence of mental health problems prior to confinement, although none of the participants indicated that they had any diagnosed mental health pathology prior to the pandemic. Another possible explanation is that, as other studies have identified [20,24,35,36], women experience work overload in addition to the responsibility of caring for their children and homes, which increases their anxiety [35,36,37]. However, this did not occur in our sample because none of the students had family burdens, such as caring for children.

In our sample, we observed a high incidence of anxiety and depression during confinement. Six months later, the participants reported more normalized levels of anxiety and depression. This is consistent with population studies [21,22,23] and studies of Italian university students [38] and Spanish health professionals [24]. However, the longitudinal study [22] that included the general Spanish population showed that although depression levels had decreased, they did not return to pre-pandemic levels.

Resilience is the ability to face adversities and recover from them. Different studies have identified resilience as a protective factor against the development of post-traumatic stress disorders [39], mental health problems [40], and burnout [41]. Bonanno (2004) [39] found that in emergency situations, people make use of their internal resources to maintain their mental health. Resilience is related to less anxiety and depression, less psychological distress, and increased well-being [18,20,39]. In our study, people who scored high in resilience had a higher quality of life score and a lower prevalence of depression and post-traumatic stress disorders. Resilience can help people cope with adverse situations and is a great resource for mitigating anxiety symptoms and improving well-being [34].

Other studies [22,24,28] have found differences in resilience according to age and gender, with males presenting higher resilience scores. In our study, we did not find such differences. Resilience can be reinforced by experience and education; in fact, a study of nursing students in Israel [34] showed that students in advanced courses showed more resilience, possibly because their experience with clinical practice in real-life settings improved their ability to mitigate stress.

Previous studies have shown that rotating shifts are related to a high risk of post-traumatic stress, depression, and anxiety [18,41]. In our study, variables such as having worked in units with COVID patients, on rotating shifts, and with low levels of experience did not show any relationship with any psychological variable. However, the professionals in our sample had only worked as nurses for 6 months, which may explain the lack of significant relationships between these factors.

To our knowledge, this is the first longitudinal study conducted in Spain that evaluates the evolution of the mental health of nursing students over two periods: from the students’ last semester of study to 6 months later, after they had graduated and were working as nurses during the pandemic.

This study is not exempt from limitations—data collection took place online, the response rate was low, and it is possible that the people who responded were those who had less anxiety and more resilience. Because the study was conducted online, we cannot determine the real response rate since we do not know how many members of the population the survey reached. The data are self-reported, and resilience, post-traumatic stress, and satisfaction with life were not measured at T0. The study design did not allow for causality to be established among the different variables. Nonetheless, this study highlights that the mental health of fourth-year nursing students was altered during the confinement period of the pandemic, although anxiety and depression decreased 6 months later, when the participants had graduated and were working as nurses. Universities and health services must make plans to provide mental health services to protect and help health professionals. It would be interesting to continue studying the evolution of the mental health of these professionals, as it has been 2 years since the pandemic began and there have since been more than six waves of the pandemic, each characterized by high mortality and workload and signs of mental fatigue, depression, etc.

### Relevance to Clinical Practice

This study highlights the importance of resilience in protecting against problems with anxiety, depression, and post-traumatic stress disorders and improving quality of life. More than two years have passed since the onset of the COVID-19 pandemic and healthcare professionals have suffered the most from the negative impacts on their psychological health. However, few health policies have been implemented to reduce this problem. Therefore, work conditions should be improved and strategic lines of mental health support should be established by universities and health services to prevent mental pathologies. Teaching strategies that increase resilience can help health professionals cope with future pandemic waves or emergency situations.

## 5. Conclusions

The COVID-19 pandemic has had a negative impact on the mental health of nursing students in Spain. This study shows the evolution of anxiety and depression levels of nursing students from the first period of confinement to 6 months later. During the period of confinement in the first wave of the pandemic, students scored high on the anxiety and depression scales. After 6 months, their anxiety and depression levels had decreased.

Resilience is related to a better quality of life and less anxiety and depression, and it may be a protective factor for mental health. Unlike other studies, no relationship was found with gender, age, and work shifts, which was observed in health professionals with more work experience. Universities and, subsequently, health services should increase interventions to improve and promote training in mental health and resilience.

## Figures and Tables

**Table 1 ijerph-19-13100-t001:** Responses to the different scales during the post-test phase (December 2020) (by analyzed total responses and by gender).

	Men *n* (%)	Women *n* (%)	Total *n* (%)	*p*
Post-traumatic Stress				0.12 ^(1)^
No	6 (60%)	64 (81.7%)	73 (79.3%)	
Yes	4 (40%)	15 (18.3%)	19 (20.7%)	
Anxiety				0.658 ^(1)^
No	9 (90%)	72 (87.8%)	81 (88%)	
Yes	1 (10%)	10 (12.2%)	11 (12%)	
Depression				0.057 ^(1)^
No	8 (80%)	80 (97.6%)	88 (95.7%)	
Yes	2 (20%)	2 (2.4%)	4 (4.3%)	
Overall LISAT				0.164 ^(1)^
Low	1 (10%)	1 (1.3%)	2 (2.2%)	
Medium	5 (50%)	33 (41.8%)	38 (42.7%)	
High	4 (40%)	45 (57%)	49 (55.1%)	
Social LISAT				0.011 *^(1)^
Low	1 (10%)	0	1 (1.1%)	
Medium	5 (50%)	29 (36.7%)	34 (38.2%)	
High	4 (40%)	50 (63.3%)	54 (60.7%)	
Sexual LISAT				0.89 ^(1)^
Low	3 (42.9%)	23 (33.8%)	26 (34.7%)	
Medium	3 (42.9%)	33 (48.5%)	36 (48%)	
High	1 (14.3%)	12 (17.6%)	13 (17.3%)	
Financial LISAT				0.626 ^(1)^
Low	4 (40%)	20 (25.6%)	24 (27.3%)	
Medium	5 (50%)	47 (60.3%)	52 (59.1%)	
High	1 (10%)	11 (14.1%)	12 (13.6%)	
Resilience	26.3 (± 7.54)	27.12 (± 8.39)	27.03 (± 8.27)	0.656 ^(2)^

* *p* < 0.05; LISAT, satisfaction with life; ^(1)^ significant value of χ^2^ test; ^(2)^ significant value of Mann–Whitney U test; anxiety > 11; depression > 11; post-traumatic stress > 12; overall LISAT (good (36–48), medium (21–35), and low (20–8)); social LISAT (good (18–24), medium (11–17), low (4–109)); sexual and financial LISATs (good (10–12), medium (4–9), and low (4–8)).

**Table 2 ijerph-19-13100-t002:** T0–T1 mean anxiety and depression scores.

	Pre-Test (May 2020)	Post-Test (December 2020)	*p*
	*n* (SD)	*n* (SD)	
Anxiety	8.36 (± 4.11)	6.83 (± 3.53)	< 0.001 ^(1)^
Depression	5.77 (± 3.52)	4.24 (± 3.36)	< 0.001 ^(1)^
	*n* (%)	*n* (%)	
Anxiety			< 0.001 ^(2)^
No anxiety (<7)	41 (44.6%)	54 (58.7%)	
Possible anxiety (8–11)	21 (22.8%)	27 (29.3%)	
Anxiety (>11)	30 (32.6%)	11 (12%)	
Depression			< 0.001 ^(2)^
No depression (<7)	61 (63.3%)	76 (82.6%)	
Possible depression (7–11)	18 (19.6%)	12 (13%)	
Depression (>11)	13 (14.1%)	4 (4.3%)	

SD, standard deviation; ^(1)^ significant value of paired Wilcoxon test; ^(2)^ significant value of χ^2^ test.

**Table 3 ijerph-19-13100-t003:** Correlations between anxiety, depression, and post-traumatic stress and satisfaction with life.

	Simple Correlation					Partial Correlation				
	Anxiety	Depression	LISAT Total	Resilience	Post-Traumatic Stress	Anxiety	Depression	LISAT Total	Resilience	Post-Traumatic Stress
Anxiety	-	0.647 **	−0.159	−0.250	0.457 **	-	0.650 **	0.023	−0.097	0.440 *
Depression		-	−0.314 *	−0.288 *	0.546 **		-	−0.073	−0.126	0.508 *
Overall LISAT			-	0.664 **	−0.409 **			-	0.805 **	−0.264 *
Resilience				-	−0.333 *				-	−0.269 *
Post-traumatic stress					-					-

Data are presented in the correlation coefficient Rho. * *p* < 0.05; ** *p* < 0.001. Abbreviations: LISAT: life satisfaction. Adjusted for sex.

## Data Availability

The data that support the findings of this study are available from the corresponding author, upon reasonable request.

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
