# Peer review of "Longitudinal Study of the Mental Health, Resilience, and Post-Traumatic Stress of Senior Nursing Students to Nursing Graduates during the COVID-19 Pandemic"

_ijerph, 2022, doi:10.3390/ijerph192013100_

Round 1

Reviewer 1 Report (New Reviewer)

Overall an interesting work. The manuscript is of interest to readers. The results of a longitudinal study are very important. I would like the authors to gather repeated information from a larger number of respondents. My comment can be taken as a suggestion. It does not detract from the work done by the authors. Authors should proofread the text again. There are technical errors. For example: Line 40 - missing a square bracket, line 260 - indicate table 3! Line 68 - you should put a space between 2020 and [11]. For 3 or more literature sources, choose a specific layout. For example, [18-20] or [18,19,20].

Author Response

We appreciate very much your constructive comments, helpful information and your time. Thanks to this review, our manuscript was substantially improved. Responses to your comments are written in bold.

Reviewer 1

Overall an interesting work. The manuscript is of interest to readers. The results of a longitudinal study are very important. I would like the authors to gather repeated information from a larger number of respondents. My comment can be taken as a suggestion. It does not detract from the work done by the authors. Authors should proofread the text again. There are technical errors. For example: Line 40 - missing a square bracket, line 260 - indicate table 3! Line 68 - you should put a space between 2020 and [11]. For 3 or more literature sources, choose a specific layout. For example, [18-20] or [18,19,20].

Authors: Thank you very much for your suggestions, we have changed technical errors.  Unfortunately, we cannot repeat the information from a larger number of respondents. In the next researches we will try to increase the responses.

Reviewer 2 Report (New Reviewer)

I think this study is a very meaningful longitudinal study comparing the psychological variables of nursing students who experienced the COVID-19 pandemic before and after graduation.

The following opinions are suggested to improve the quality of the paper.

1.     Describe Reliability for each scale

2.     Please clearly state the scale that applies separately to the T0 and T1 stages. Consider presenting the data collection procedure as a figure.

3.     “with less depression (2 p = 0.020)” : In addition to the p-value, please provide the chi statistic

4.     Table 1: Indicate cutoff scores and categories for each variable as footnotes in the table.

5.     Table 2: Anxiety has been duplicated, so delete it.

6.     Table 3: Add explanation for the meaning of partial correlation analysis results.

7.     Typo correction: “such as those by Patelarou el al (2021)”

8.     Discussion description should focus more on the results of this study.
ex) “
In our study, people who scored high in resilience had a higher quality of life score and lower prevalence of depression, anxiety and posttraumatic stress disorders.”

è  As a result of the correlation analysis in this study, there was no significant correlation between anxiety and resilience.

Author Response

We appreciate very much your constructive comments, helpful information and your time. Thanks to this review, our manuscript was substantially improved. Responses to your comments are written in bold.

Reviewer 2

I think this study is a very meaningful longitudinal study comparing the psychological variables of nursing students who experienced the COVID-19 pandemic before and after graduation.

The following opinions are suggested to improve the quality of the paper.

  1. Describe Reliability for each scale

Authors: Thank you for your suggestion. We have added it.

  1. Please clearly state the scale that applies separately to the T0 and T1 stages. Consider presenting the data collection procedure as a figure.

Authors: Thank for your suggestions, we have added in the text.

  1. “with less depression (ꭕ2p = 0.020)” : In addition to the p-value, please provide the chi statistic

Authors: Thank you for your suggestion. We have added it.

  1. Table 1: Indicate cutoff scores and categories for each variable as footnotes in the table.

Authors: Than you for your suggestions. We have added it.

  1. Table 2: Anxiety has been duplicated, so delete it.

Authors: We have done!

  1. Table 3: Add explanation for the meaning of partial correlation analysis results.

Authors: Thank for your suggestions, we have added in the text.

  1. Typo correction: “such as those by Patelarou elal (2021)”

     Authors: We have done!

  1. Discussion description should focus more on the results of this study.
    ex) “In our study, people who scored high in resilience had a higher quality of life score and lower prevalence of depression, anxiety and posttraumatic stress disorders.”

è  As a result of the correlation analysis in this study, there was no significant correlation between anxiety and resilience.

 Authors: Thank you very much, it was a mistake. We have corrected it.

This manuscript is a resubmission of an earlier submission. The following is a list of the peer review reports and author responses from that submission.

Round 1

Reviewer 1 Report

Thank you for submitting your article to this journal. I read it with interest. I think the theme of this study is important, however, this article has a lot of fundamental challenges, such as lack of consistency throughout the text. Referring below comments, please improve your article with keep consistency.

1. Title 

The title does not match the content of the text. In the results section, it seems that the main findings were the sex differences of the dependent variables and the correlation coefficients between the resilience and the other dependent variables. Please keep consistency with title and the text.

2. Abstract 

Please add the number of the participants with response rate. Additionally, please improve the whole contents referring to the following comments. 

3. Introduction

The logic, which leads to the purpose of this study, is inappropriate from line 83 to 98. Until line 82, the authors mentioned about the COVID-19 pandemic and the factors of anxiety of university students due to the pandemic. However, from line 83, the authors abruptly mentioned that they focus on the psychological health of nursing students. The authors should have mentioned which kinds of psychological health they aim to examine with the rational reasons showing the evidence. Also, the authors did not mention about sex differences of their psychological health nor the resilience which is the capacity to recover quickly from difficulties in this section. However, these results were shown in the results and discussion sections as if they are main findings. Please keep consistency throughout the text.

 4. Materials and Methods:

1) In these kind of longitudinal studies, researchers should match individual variables from Time 0 to Time 1, and conduct the Wilcoxson's signed rank test to examine the change. However, the authors conducted panel surveys and conducted analysis both Time 0 and Time 1 separately. There were no clear descriptions about the rational reasons to choose the analysis. Why did the authors examine the correlations coefficients between each dependent variable? Why did the authors examine the difference of each score of dependent variables only according to sex? Please choose appropriate statistical methods for the purpose of this study.

2) There are some independent variables such as age, smoking status, and number of cigarettes, etc, of which were not shown in results and not mentioned in the discussion section at all. Please add appropriate explanations.

5. Results

There are a lot of inappropriate descriptions. For example, inappropriate data (e.g., in each table, confusing writing of commas and decimal points, the number of decimal places is not unified), and regarding the contents of each table, there are many repetitive descriptions in the text. Also, generally, if the authors show tables,  they should minimize repetitive descriptions of tables in the text .

This study aims to analyze the psychological health of students who were in their last year of the nursing degree during the COVID-19 pandemic. However, the main findings were the sex differences regarding the dependent variables, as well as the correlation coefficients between the resilience, which is the capacity to recover quickly from difficulties, and the other dependent variables. Please keep consistency throughout the text.

6. Discussion

In the discussion section, there are a lot of challenges which were not based on the results that appear abruptly. For example, L280-L282, in spite of that the authors did not show any variables regarding “having received health”, ”work locations or work shift”, the authors then suddenly discuss it. In addition, when the authors discuss the relationships between two variables based on the correlation coefficients, they need to understand the meaning of the score level (weak correlation, moderate correlation, strong correlation).

 Please refer to the above-mentioned comments and keep consistency from the introduction section to the discussion section. Additionally, please improve the limitation and the conclusion section after improving the whole text in the appropriate way.

Author Response

Reply to reviewer 1.

Authors: We appreciate very much your constructive comments, helpful information and your time. Thanks to this review, our manuscript was substantially improved. Responses to your comments are written in red.

R1 comments

Thank you for submitting your article to this journal. I read it with interest. I think the theme of this study is important, however, this article has a lot of fundamental challenges, such as lack of consistency throughout the text. Referring below comments, please improve your article with keep consistency.

  1. Title

The title does not match the content of the text. In the results section, it seems that the main findings were the sex differences of the dependent variables and the correlation coefficients between the resilience and the other dependent variables. Please keep consistency with title and the text.

  • Authors: thank you for your suggestions. We have changed the title:Longitudinal study of mental health, resilience, and post-traumatic stress of senior nursing students to nursing graduates during the Covid-19 pandemic.
  1. Abstract

Please add the number of the participants with response rate. Additionally, please improve the whole contents referring to the following comments.

  • Authors: Thank you, we have added in the abstracts the number of the participants with response rate.
  1. Introduction

The logic, which leads to the purpose of this study, is inappropriate from line 83 to 98. Until line 82, the authors mentioned about the COVID-19 pandemic and the factors of anxiety of university students due to the pandemic. However, from line 83, the authors abruptly mentioned that they focus on the psychological health of nursing students. The authors should have mentioned which kinds of psychological health they aim to examine with the rational reasons showing the evidence. Also, the authors did not mention about sex differences of their psychological health nor the resilience which is the capacity to recover quickly from difficulties in this section. However, these results were shown in the results and discussion sections as if they are main findings. Please keep consistency throughout the text.

  • Authors: Thank you for your suggestions, we have modified the introduction.
  1. Materials and Methods:

1) In these kinds of longitudinal studies, researchers should match individual variables from Time 0 to Time 1, and conduct the Wilcoxson's signed rank test to examine the change. However, the authors conducted panel surveys and conducted analysis both Time 0 and Time 1 separately. There were no clear descriptions about the rational reasons to choose the analysis. Why did the authors examine the correlations coefficients between each dependent variable? Why did the authors examine the difference of each score of dependent variables only according to sex? Please choose appropriate statistical methods for the purpose of this study.

  • Authors: Thank you very much for your suggestion.
  • We have added the Wilcoxon test to know the changes produced between two times (T0-T1).
  • Our objective is to know the psychological impact of covid on nursing students and the changes that occur over time when they are already nurses. In addition, we were interested in knowing the impact on women, since in studies with individuals in general, women were associated with a greater psychological impact (Whang 2020b, Gonzalez 2020). In studies with nurses, female nurses show worse mental health during the pandemic and higher levels of emotional exhaustion (Peñacoba et al., 2021; Luceño-Moreno et al., 2021), considering the high percentage of women working as nurses in the healthcare system it is interesting to know. Therefore, for these reasons, we examined the scoring of the dependent variables as a function of sex.

-González-Sanguino, C., Ausín, B., Castellanos, M. A., Saiz, J., & Muñoz, M. Mental health consequences of the Covid-19 outbreak in Spain. A longitudinal study of the alarm situation and return to the new normality. Progress in Neu-ro-Psychopharmacology and Biological Psychiatry, 2021,  107, 110219

-Peñacoba, C., Velasco, L., Catalá, P., Gil‐Almagro, F., García‐Hedrera, F. J., & Carmona‐Monge, F. J. Resilience and anxiety among intensive care unit professionals during the COVID‐19 pandemic. Nursing in Critical Care, 2021, 26(6), 501-509

-Luceño-Moreno, L., Talavera-Velasco, B., Vázquez-Estévez, D., & Martín-García, J. Mental Health, Burnout and Resilience in Healthcare Professionals After the First Wave of COVID-19 Pandemic in Spain: A Longitudinal Study. Journal of Occupational and Environmental Medicine, 2021

-Wan, Z., Lian, M., Ma, H., Cai, Z., & Xianyu, Y. Factors associated with burnout among Chinese nurses during COVID-19 epidemic: A cross-sectional study. BMC Nursing, 2022, 21(1), 1–8. https://doi.org/10. 1186/s12912-022-00831-3

2) There are some independent variables such as age, smoking status, and number of cigarettes, etc, of which were not shown in results and not mentioned in the discussion section at all. Please add appropriate explanations.

Authors: Thank you for your suggestions.

We have seen that age has not relationship with psychological impact because most of the students were the same age. During the first administration (T0), which took place during the confinement period, 296 students (response rate = 68.83%) aged 21-54 years (M = 23.4: SD = 4.79) answered the questionnaires in full. Only 12 (13%) persons were older than 30 years, and we found no statistical differences by age. We think that the small size of the older students could be the reason because in our study there are not psychological differences versus other studies with nurses with more age variability.  In the line 289-290 (discussion section) we say this: “In our study, this was not the case, possibly because the students in our study were of similar ages and had not previously worked in the health system; they only had experience in clinical practice”.

With respect to tabaco, only 13 people smoked, and the number of cigarettes smoked did not change (T0-T1).

We have added a comment in the results and discussion.

  1. Results

There are a lot of inappropriate descriptions. For example, inappropriate data (e.g., in each table, confusing writing of commas and decimal points, the number of decimal places is not unified), and regarding the contents of each table, there are many repetitive descriptions in the text. Also, generally, if the authors show tables,  they should minimize repetitive descriptions of tables in the text .

Authors: Thank you for your suggestion. We have revised the tables and data; we have changed commas for decimal points. At the editor's suggestion we have removed table 1. We have tried to minimize repetitive descriptions.

  1. Discussion

In the discussion section, there are a lot of challenges which were not based on the results that appear abruptly. For example, L280-L282, in spite of that the authors did not show any variables regarding “having received health”, ”work locations or work shift”, the authors then suddenly discuss it. In addition, when the authors discuss the relationships between two variables based on the correlation coefficients, they need to understand the meaning of the score level (weak correlation, moderate correlation, strong correlation).

Authors: Thank you very much for your suggestions. In the results section, in line 215 we show that 47,8% of the participants had provide care assistance to the public health.  In the discussion there is a translation error, we meant to say, “had provide care assistance to the public health”, we changed it. In results we have added data on shift and place work (line 218-219)

 Please refer to the above-mentioned comments and keep consistency from the introduction section to the discussion section. Additionally, please improve the limitation and the conclusion section after improving the whole text in the appropriate way.

Authors: Done

Reviewer 2 Report

Thank you for the opportunity to review this study entitled “Evolution of mental health of final year nursing students to nursing graduates during the Covid-19 pandemic: A cohort study.” (ijerph-1855264).

The research was a longitudinal prospective study focused on the psychological effect of the COVID-19 pandemic, by investigating changes in the psychological health of students who were in the final year of their nursing degree during the pandemic and later served as nursing professionals in hospitals.

In my opinion, the research topic is relevant, and the study is interesting. Parallelly, there are some issues that need to be addressed before the paper will be suitable for publication.

1.     Abstract: the information about the sample should be deepened (N? Mean age and SD? Percentage of men and women?) to provide a clear picture of what will be presented in the paper.

2.     Abstract: Please remove headings, according to the IJERPH guidelines.

3.     Introduction: In my opinion, it would be good to refer to trend or longitudinal studies, if any. Since the authors frame this study considering the impact that COVID-19 has on a psychological level, I suggest some research to propose a comprehensive framework in the introduction, which should be supplemented with further literature search by the authors:

-       Hyland et al., 2021; doi: 10.1016/j.psychres.2021.113905.

-       Gori & Topino, 2021; doi: 10.3390/ijerph18115651

-       Wang et al., 2020; doi: 10.1016/j.bbi.2020.04.028

-       To find the suggested articles, the authors can use this source: https://www.doi.org/

4.     The Materials and Methods section looks chaotic. I suggest combining the "Design", "Reference population / study area", "Inclusion criteria", "Exclusion criteria", and "Sample size and sampling" parts into a single section.

5.     Should “Relevance to clinical practice” at the end of the discussion be a heading?

6.     Both the “Relevance to clinical practice” and the “Conclusions” sections should be enriched.

Best wishes

Author Response

Reply to reviewer 2

 We appreciate very much your constructive comments, helpful information and your time. Thanks to this review, our manuscript was substantially improved. Responses to your comments are written in red.

Comment of R2

Thank you for the opportunity to review this study entitled “Evolution of mental health of final year nursing students to nursing graduates during the Covid-19 pandemic: A cohort study.” (ijerph-1855264).The research was a longitudinal prospective study focused on the psychological effect of the COVID-19 pandemic, by investigating changes in the psychological health of students who were in the final year of their nursing degree during the pandemic and later served as nursing professionals in hospitals.In my opinion, the research topic is relevant, and the study is interesting. Parallelly, there are some issues that need to be addressed before the paper will be suitable for publication.

  1. Abstract: the information about the sample should be deepened (N? Mean age and SD? Percentage of men and women?) to provide a clear picture of what will be presented in the paper.

Authors: Thank you for your suggestions, we added data in the abstract.

  1. Abstract: Please remove headings, according to the IJERPH guidelines.

Author: Thank you, we done it.

  1. Introduction: In my opinion, it would be good to refer to trend or longitudinal studies, if any. Since the authors frame this study considering the impact that COVID-19 has on a psychological level, I suggest some research to propose a comprehensive framework in the introduction, which should be supplemented with further literature search by the authors:

-       Hyland et al., 2021; doi: 10.1016/j.psychres.2021.113905.-       Gori & Topino, 2021; doi: 10.3390/ijerph18115651-       Wang et al., 2020; doi: 10.1016/j.bbi.2020.04.028-       To find the suggested articles, the authors can use this source: https://www.doi.org/

Authors: Thank you for your suggestions, we have rewritten the introduction section.

  1. The Materials and Methods section looks chaotic. I suggest combining the "Design", "Reference population / study area", "Inclusion criteria", "Exclusion criteria", and "Sample size and sampling" parts into a single section.

Authors: Thank you for your suggestions, we done it.

  1. Should “Relevance to clinical practice” at the end of the discussion be a heading?

Authors: Thank you. Done it.

  1. Both the “Relevance to clinical practice” and the “Conclusions” sections should be enriched.

Authors: Thank you. Done it.

Round 2

Reviewer 1 Report

Although you changed the text to improve this article, there are still a lot of fundamental challenges. Referring below comments, please improve your article with keep consistency.

1. Although I pointed out that there are a lot of inappropriate descriptions, there are still a lot of such description. For example, the number of decimal places is not unified (L24-25, L214, L134, Table 2, Table 4, etc), inappropriate notation of data (e.g., L241-245), inappropriate and meaningless tables so that I can not understand what the authors try to show (e.g.,Table 3, Table 4), incomplete sentences (e.g., L110, L254, L361, etc), and regarding the contents of each table, there are many repetitive descriptions in the text. Also, generally, if the authors show tables, they should minimize repetitive descriptions of tables in the text. Please improve them in the text and tables.

2. Materials and Methods, and Results

(1) This article is lack of many fundamental manners, and still has a lot of critical challenges as research article. For example, a partial correlation coefficient is a measure of the linear dependence of a pair of random variables from a collection of random variables in the case where the influence of the remaining variables is eliminated. However, the authors did not mention about the procedures about the model for analysis appropriately. The description from L 197 to L198 need more information. Please add information and improve the text.

(2) I doubt veracity of the description from L192 to L 194 which mentioned that the data fit a normal distribution. This research used several scales and the sample size was very small (n = 92). So I can not trust that description. Please show the information about the scales which showed the normal distribution. Due to this condition,I doubt veracity of the results in the tables. Please clarify that and add information.

(3) The table 3 and table 4 requires appropriate more information about the data since I can not understand what the authors tried to show in the tables. Please improve them and show appropriate tables.

(4) The authors did not improve the text and tables according to my comments last time I gave you as below. Please improve them.

<Last time comments I gave the authors>

 Results:

There are a lot of inappropriate descriptions. For example, inappropriate data (e.g., in each table, confusing writing of commas and decimal points, the number of decimal places is not unified), and regarding the contents of each table, there are many repetitive descriptions in the text. Also, generally, if the authors show tables, they should minimize repetitive descriptions of tables in the text.

3. Discussion

In the discussion section, there are still a lot of challenges which were not based on the results. For example, although there is not results about “age”, the authors discussed about age repeatedly, even in the conclusion section. Despite of the fact that this article did not use any scales for assessing work conditions and support systems, in the L367-L370, the authors discussed as if they examined such conditions using the variables. Please keep consistency throughout the text.